# Pep19 Has a Positive Effect on Insulin Sensitivity and Ameliorates Both Hepatic and Adipose Tissue Phenotype of Diet-Induced Obese Mice

**DOI:** 10.3390/ijms23084082

**Published:** 2022-04-07

**Authors:** Renata Silvério, Robson Barth, Andrea S. Heimann, Patrícia Reckziegel, Gustavo J. dos Santos, Silvana Y. Romero-Zerbo, Francisco J. Bermúdez-Silva, Alex Rafacho, Emer S. Ferro

**Affiliations:** 1Graduate Program in Pharmacology, Federal University of Santa Catarina (UFSC), Florianópolis 88040-900, Brazil; resilveriodeluca@gmail.com; 2Laboratory of Investigation in Chronic Diseases, Department of Physiological Sciences, Federal University of Santa Catarina (UFSC), Florianópolis 88040-900, Brazil; robsom.barth@gmail.com (R.B.); gustavo.js@ufsc.br (G.J.d.S.); 3Multicenter Graduate Program in Physiological Sciences, Federal University of Santa Catarina (UFSC), Florianópolis 88040-900, Brazil; 4Proteimax BioTechnology Israel LTD, 4 Duvdevan Street, Pardes Hana, Haifa 3708973, Israel; andrea@proteimax.com; 5Department of Pharmacology, Biomedical Science Institute, University of São Paulo (USP), São Paulo 05508-000, Brazil; reckziegel.patricia@gmail.com; 6Instituto de Investigación Biomédica de Málaga-IBIMA, UGC Endocrinología y Nutrición Hospital Regional Universitario de Málaga, Universidad de Málaga, 29009 Málaga, Spain; yanina.romero@ibima.eu (S.Y.R.-Z.); javier.bermudez@ibima.eu (F.J.B.-S.); 7Biomedical Research Center for Diabetes and Associated Metabolic Diseases (CIBERDEM), 28029 Madrid, Spain

**Keywords:** bioactive peptides, orally active peptides, intracellular peptides, overweight, obesity, metabolism

## Abstract

Peptide DIIADDEPLT (Pep19) has been previously suggested to improve metabolic parameters, without adverse central nervous system effects, in a murine model of diet-induced obesity. Here, we aimed to further evaluate whether Pep19 oral administration has anti-obesogenic effects, in a well-established high-fat diet-induced obesity model. Male Swiss mice, fed either a standard diet (SD) or high-fat diet (HFD), were orally administrated for 30 consecutive days, once a day, with saline vehicle or Pep19 (1 mg/kg). Next, several metabolic, morphological, and behavioral parameters were evaluated. Oral administration of Pep19 attenuated HFD body-weight gain, reduced in approximately 40% the absolute mass of the endocrine pancreas, and improved the relationship between circulating insulin and peripheral insulin sensitivity. Pep19 treatment of HFD-fed mice attenuated liver inflammation, hepatic fat distribution and accumulation, and lowered plasma alanine aminotransferase activity. The inguinal fat depot from the SD group treated with Pep19 showed multilocular brown-fat-like cells and increased mRNA expression of uncoupling protein 1 (UCP1), suggesting browning on inguinal white adipose cells. Morphological analysis of brown adipose tissue (BAT) from HFD mice showed the presence of larger white-like unilocular cells, compared to BAT from SD, Pep19-treated SD or HFD mice. Pep19 treatment produced no alterations in mice behavior. Oral administration of Pep19 ameliorates some metabolic traits altered by diet-induced obesity in a Swiss mice model.

## 1. Introduction

Multifactorial causes are responsible for overweight and obesity, including genetic, behavioral, environmental, physiological, social, and cultural factors that result in energy imbalance and promote excessive fat deposition [1]. Obesity is commonly associated with an increased risk of developing type 2 diabetes, hypertension, and/or cancer, resulting in lower life expectancy [1,2,3]. The prevalence of overweight or obesity worldwide continues to grow significantly, with projections to reach around 80% of the world population by 2030 [4]. Populational mitigation of weight excess by the introduction of a healthy lifestyle is undoubtedly necessary; however, adherence to these recommendations is very poor and, hence, pharmacological interventions are necessary.

The endocannabinoid system (i.e., endocannabinoids, receptors, and metabolizing enzymes) plays an important role in the control of energy metabolism, due to its regulatory actions on appetite, lipolysis, and energy expenditure [5,6]. Accordingly, people with obesity may exhibit an overactivity of the endocannabinoid system [7,8]. Several animal models of obesity can reproduce human overweight and obesity, and there are many protocols, such as the high-fat diet (HFD), that lead to excess fat deposition [9]. Indeed, the lack of effect of SR141716 (rimonabant), a selective CB1-R inverse agonist, in CB1-R knockout HFD-fed mice, further supports the key function played by these receptors in the development of diet-induced obesity in humans and rodents [10,11,12]. Clinical trials with rimonabant, in obese and diabetic individuals, results in a reduction in body weight and improvement in peripheral insulin sensitivity [13,14,15,16]. On the other hand, stimulation of CB1-R with the agonist WIN 55212.2 reduced the thermogenesis of brown adipocytes, favoring a positive energy balance and weight gain [17]. Despite the positive effects on metabolism, the first generation of CB1-R-blocking drugs, such as rimonabant, was withdrawn from the market due to psychiatric adverse effects, such as depression and anxiety [18,19,20]. Nowadays, new strategies have been investigated to abrogate or mitigate the central psychiatric side effects of the CB1-R signaling pathway, including the development of CB1-R antagonists that do not cross the blood–brain barrier, neutral antagonists, and allosteric modulators [21,22]. For example, Chronic peripheral blockade of type 1 cannabinoid receptors (CB1-R) in obese mice, using the strictly peripheral CB1-R antagonist AM6545, induces weight loss and an improvement in dyslipidemia [23].

Orally active peptides have been previously described [24,25,26,27,28], including hemopressin and derived peptides [27,29,30,31,32,33,34,35,36,37], Ric4 [38], peptides C111/C112 from bonito liver [39], ubiquitin ligase inhibitory pentapeptide [40], in addition to milk peptides [41,42,43,44,45]. Moreover, previous study has shown that natural peptide DIIADDEPLT (Pep19) oral administration to a Wistar rat model of diet-induced obesity, which consisted of administrating a 20% sucrose solution, instead of water, for 13 consecutive weeks, improves metabolic parameters without adverse central nervous system effects [21]. Pep19 lacks central nervous system activity, as suggested by the absence of brain c-Fos expression, induction of the cannabinoid tetrad, and depressive- and anxiety-like behaviors [21]. The expression of uncoupling protein 1 (UCP1) in specific cells of the inguinal adipose tissue was significantly increased following oral administration of Pep19 to diet-induced obese Wistar rats [21]. Moreover, AM251, a CB1-R antagonist, blocked UCP1 expression induced by Pep19 in 3T3-L1-differentiated cells [21]. Pep19 activates both extracellular signal-regulated kinases (pERK1/2) and protein kinase B (AKT) signaling pathways [21]. Together, these data suggested that Pep19 could improve metabolic parameters, activating UCP1 to induce browning of the white adipose tissue [21]. In the present report, we investigated whether oral administration of Pep19 could improve metabolic parameters and exert protection of the obesogenic outcomes induced by an HFD in a Swiss mice model. The present data suggest that animals fed a standard diet (SD) that were orally administrated with Pep19, presented multilocular brown-fat-like cells in the inguinal white adipose tissue, corroborating previous suggestions that Pep19 can induce the conversion of white to brown fat cells [21]. Moreover, oral administration of Pep19, to animals fed an HFD, exerted a beneficial impact on body-weight gain, insulin sensitivity, ameliorated hepatic steatosis and inflammation, and prevented brown adipose tissue (BAT) from presenting large numbers of white adipocyte-like unilocular cells. Further studies shall be conducted to deeply investigate the molecular mechanisms related to the oral bioavailability and metabolic improvements related to Pep19 administration.

## 2. Materials and Methods

### 2.1. Animals

The present experimental study was approved by the Committee for Ethics in Animal Experimentation of the Federal University of Santa Catarina (Protocol No. 5560080118, from 5 October 2018), in accordance with the Brazilian College for Animal Experimentation (COBEA). This study strictly followed animal care and health controls for all animals used herein, as standardized by local and international ethics in animal experimentation counsels. These strict animal care and health controls conditions were carried out at all times, both before and during the experiments. 

A total of 50 male SWR/J outbred Swiss mice (6 weeks old) were used in this study. The animals were supplied by the Federal University of Santa Catarina’s Animal Breeding Center and transferred to a laboratory animal house facility (5 mice per cage) where they were kept at 24 ± 1 °C on a 12-h light–dark cycle (lights on 06:00–18:00). The cage leadership was defined with the arrival and random separation of animals during the acclimation period, while fight behaviors were not observed. All mice had *ad libitum* access to food (commercial standard chow, Nuvilab CR-1; Nuvital, Colombo, PR, Brazil) and filtered tap water.

### 2.2. Experimental Design and Groups

Swiss mice were chosen as a model of HFD-induced obesity based on previous studies from our group [46,47]. In these previous studies, Swiss mice strand was shown to respond efficiently to the HFD, developing obesity features (e.g., elevated body mass and adiposity, glucose intolerance and insulin resistance) [46,47]. Thus, mice were acclimatized for 4 weeks before being randomly assigned to one of the four experimental groups (*n* = 10–15 per group). Animals were assigned to receive, for 30 consecutive days, either a standard diet (SD) or an HFD (Table 1).

Half of the animals from each group (SD and HFD) received concomitant administration of either saline vehicle (equivalent to 4 mL/kg) or saline-diluted Pep19 (1 mg/kg; Proteimax Biotechnology LTDA, São Paulo, SP, Brazil; peptide synthesis was performed as previously described [48]), once daily, between 08:00 and 9:00 a.m.; Pep19 was always administrated by the same person, who conducted the oral gavage administration of Pep19 using 18-gauge feeding tubes about 1.5 inches in length with a rounded tip. The four experimental groups were labeled: SD, HFD, SD treated with Pep19 (SD-P), and HFD treated with Pep19 (HFD-P). The saline vehicle and Pep19 volume/dose were adjusted every two days according to mice body weight. 

### 2.3. Body Mass and Food Intake

Body mass and food intake were determined every 2 days. A digital electronic balance (TECNAL, São Paulo, Brazil) was used to determine body mass. Percentage (%) of body mass variation was determined as the following: [(final body mass-initial body mass)/initial body mass] × 100. Every 48 h individual food intake was determined measuring the difference in the weight of the chow offered and the weight of the chow remaining in the cage, divided by the number of animals from each cage. Feed efficiency was given by the body mass (g) gain divided by the total caloric consumption (Kcal) [46].

### 2.4. Glucose Tolerance and Insulin Sensitivity Assessment

During the last week of the experimental period, insulin sensitivity and glucose tolerance tests were assessed by intraperitoneal insulin (ipITT) and glucose tolerance test (ipGTT), respectively; these experiments started around 02:00 p.m. The ipITT was performed on day 26 of treatment. For this, mice were fasted for 2 h (to mitigate the influence of any previous food ingestion on baseline blood glucose values) and then the tail tip was cut (0.5 mm; a 2% lidocaine spray was applied between 5 to 10 min before) for blood sampling. The first blood drop was discarded and the second one was used for blood glucose determination (baseline, time 0) using a glucometer (Accu-Check Performa; Roche Diagnostics, Mannheim, Germany). Then 0.75 IU/kg body mass of insulin (regular Humulin^®^, Lily, Indianapolis, IN, USA) was intraperitoneally (ip) administered, and additional samples were collected at 15 and 30 min for blood glucose measurement. Blood glucose values were expressed as a percentage of initial blood glucose value and the constant of glucose disappearance (Kitt) was calculated as previously described [49]. The ipGTT was performed on day 28 of treatment in 6 h fasted mice. Blood was sampled from the tail tips as described before (baseline, time 0) and, immediately, an ip 50% glucose solution (1.5 g/kg) was administered with additional samples being collected at 15, 30, 60, and 120 min.

### 2.5. Euthanasia and Tissue Harvesting

On the 30th day of treatment, mice were fasted for 6 h, anesthetized with xylazine/ketamine (10/100 mg/Kg ip), and euthanized by decapitation in a rodent guillotine. Trunk blood was collected into EDTA-containing tubes, centrifuged at 1500× *g* at room temperature for 10 min, and plasma was stored at −80 °C for later analyses of total cholesterol and triglycerides (Biotécnica, Varginha, MG, Brazil), aspartate aminotransferase (AST) and alanine aminotransferase (ALT) activities (Labtest^®^, São Paulo, SP, Brazil), and insulin (Thermo Scientific, Waltham, MA, USA, EMINS). White adipose tissue depots (epididymal, retroperitoneal, inguinal, and mesenteric), BAT (interscapular), liver, gastrocnemius muscle, and the entire pancreas were gently removed and weighed. Adiposity index was calculated as the percentage of the sum of visceral adipose depots (mesenteric, epididymal, and retroperitoneal) divided by total body weight (∑adipose fat pads/body weight) × 100. Tissues were either fixed in 4% paraformaldehyde or 10% buffered (as described hereafter) or stored at −80 °C until use.

### 2.6. Histological Analyzes

After euthanasia, samples of the inguinal and mesenteric white adipose tissue, and interscapular BAT were fixed in 4% paraformaldehyde in 0.1 M phosphate buffer, pH 7.4 for 24 h. Liver and pancreas samples were fixed in 10% buffered formalin, pH 7.4 for 24 h. Then, samples were dehydrated in increasing ethanol solutions concentration, cleared in xylene, and embedded in paraffin. Five micrometer thick sections were cut on an automated rotary microtome (RM2255, Leica Biosystems, Buffalo Grove, IL, USA) and finally stained with hematoxylin and eosin (H&E). Axio Scan slide scanner (ZEISS, Oberkochen, Germany) was used for scanning the stained sections, while determination of morphometric parameters was conducted using ImageJ software. Cell area of adipose depots was manually obtained contouring cells with intact plasma membranes (300 adipocytes counted per animal, in 10 different fields of the H&E-stained sections). The pancreatic islets’ area was obtained by manual tracing of all islets on each section (two sections per animal interspaced by 200 μm). The relative mass of the endocrine pancreas was calculated by dividing the sum of islets’ areas over the entire pancreas area × 100; the endocrine pancreas absolute mass (mg) was then estimated by multiplying their relative mass by the total pancreas mass. The representative images were captured with OLYMPUS BX41 (OLYMPUS; Tokyo, Japan). 

### 2.7. Histopathological and Biochemical Liver Analyses

The histopathological liver analyses to evaluate steatosis and inflammation were adapted from Liang et al. [50]. Briefly, severity of steatosis was graded based on the percentage of the total area affected, as follows: category 0 (<5%); category 1 (5–33%); category 2 (34–66%); category 3 (>66%). Inflammation was evaluated by counting the number of inflammatory foci per field (view size of 3.1 mm^2^), and it was scored into the following categories: normal—0 (<0.5 foci); slight—1 (0.5–1.0 foci); moderate—2 (1.0–2.0 foci); severe—3 (>2.0 foci). Both steatosis and inflammation were evaluated at 100 × magnification. These experiments were performed by a researcher blind to the different experimental groups. Determination of hepatic triacylglycerol content was carried out using the whole liver lobe according to a previous publication; the same hepatic lobe was used to collect fragments from all animals [51]. Triacylglycerol and total cholesterol content were determined on plasma collected after blood centrifugation (400× *g*, 7 min) according to the manufacturer’s instructions (Biotécnica, Varginha, MG, Brazil) [46]. Plasma alanine aminotransferase and aspartate aminotransferase were measured by using commercially available kits, according to the manufacturer’s instructions (Labtest^®^, São Paulo, SP, Brazil).

### 2.8. Real-Time PCR

Real-time polymerase chain reactions (qPCRs) experiments were performed as previously described [38] to evaluate UCP1 mRNA expression levels in the inguinal adipose tissue of animals fed either SD or HFD, who received either control vehicle or Pep19 (1 mg/kg; Proteimax Biotechnology LTDA, São Paulo, Brazil). Briefly, inguinal adipose tissues were homogenized and total RNA extracted using Trizol according to the manufacturer’s instructions (TRIzol, Life Technologies, Rockville, MD, USA). Two µg of RNA was reverse transcribed with a high-capacity cDNA reverse transcription kit (Applied Biosystems, Foster City, CA, USA, catalog number 4368814), diluted and incubated with primers previously mixed with FAST SYBR Green Master Mix (Applied Biosystems, Foster City, CA, USA, catalog number 4385612). Prism 7900 sequence detection system (Applied Biosystems, Foster City, CA, USA) was used to perform qPCRs. mRNA targets’ expression was normalized by expression of ribosomal protein L19 (Rpl19) and expressed as relative values using the 2 DDCt method [30]. Primer sequences were as follows: Rpl19 (Fwd CAATGCCAACTCCCGTCA, Rev: GTGTTTTTCCGGCAACGAG) and UCP1 (Fwd: GGCCTCTACGACTCAGTCCAT, Rev: AAGCCGGCTGAGATCTTGT). Possible alterations of Ucp1 expression levels were expressed relative to control SD group.

### 2.9. Behavioral Tests 

To evaluate whether chronic administration of Pep19 produced depressive and/or anxiety-like behaviors in obese mice, we carried out the tail suspension and plus-maze tests on the 29th day of the experimental period. Behavioral tests were conducted during the light phase (09:00–12:00) in a room with controlled temperature (23 °C, with 40%–60% humidity), with low-light intensity, as previously described [52]. These experiments were performed by a researcher blind to the different experimental groups.

### 2.10. Statistical Analysis

Two-Way ANOVA was used for time-dependent variables, and in these cases Tukey’s post-hoc test was used. For column analyses we used unpaired *t*-test for parametric variables, whereas Mann–Whitney *t*-test was used for non-parametric variables. We also performed correlation analyses with Pearson r calculation, which reports the value of the correlation coefficient (not a regression line) [53]. Correlation computes a correlation coefficient and its confidence interval. Its value ranges from −1 (perfect inverse relationship; as X goes up, Y goes down) to 1 (perfect positive relationship; as X goes up, so does Y). A value of zero means no correlation at all. When two variables vary together, there is great covariation or correlation between two variables [53]. Data were presented as mean ± standard error of the mean (SEM). Values of *p* < 0.05 were considered statistically significant in all analyses. GraphPad Prism 8 software (GraphPad Inc.; San Diego, CA, USA) was used to perform all statistical analyses.

## 3. Results

### 3.1. Pep19 Attenuates Body-Weight Gain in Mice Fed a High-Fat Diet (HFD)

Animal models of obesity can reproduce human overweight and obesity, and there are many protocols, such as the HFD used herein, which lead to excess fat deposition [9]. As expected, mice fed an HFD became obese when compared to those fed an SD, with significant differences (*p* < 0.05) in body-mass gain (Figure 1A). To evaluate the possible metabolic effects of Pep19, body-weight gain, caloric intake, adiposity index, and absolute and relative mass of tissues were analyzed. Pep19 did not influence the body-mass gain of SD animals (SD-P group), whereas in HFD-P, an attenuation of body-mass gain was statistically significant closer to the end of the experimental period (Figure 1A). The body-mass gain of individual animals was also accessed, correlating the initial and final body mass (Figure 2B). These latter data show a reduced linear correlation in the HFD-P group (Figure 2B; *r* = 0.54), compared to a positive correlation observed in the HFD group (Figure 2B; *r* = 0.85); these data further suggested that Pep19 prevented mice from gaining body mass during the experiment. Caloric intake remained similar among the groups, except for a higher caloric intake in the HFD, but not in the HFD-P group, when compared with other groups (Figure 1C, D). No differences in feed efficiency were found between HFD and HFD-P mice (Figure 1E). Adiposity index was lower in the SD-P group compared to the SD mice, whereas no differences were observed among HFD vs. HFD-P groups (Figure 1F). 

The absence of body-mass variation, caloric intake and feed efficiency, observed between SD and SD-P mice, were not followed by a lack of absolute or relative individual tissue weight (Table 2); these data suggest that individual tissue mass may not be parallel with the observed body-mass variation. Mice fed an SD administrated with Pep19 (SD-P) presented lower absolute mass of the white mesenteric adipose tissue, as well as BAT (Table 2). The relative mass of all white adipose tissues, including inguinal, epididymal, mesenteric and retroperitoneal, were decreased by Pep19 administration to mice fed an SD (Table 2). In SD-fed mice, the relative mass of BAT was also reduced by Pep19 treatment (Table 2). These data corroborate the lower adipose index of SD-P mice (Figure 1F). Moreover, SD-P mice have higher relative heart and muscle tissue weights (Table 2).

There were marked increases in fat depots weight (absolute and relative) in HFD and HFD-P groups compared, respectively, to SD or SD-P (Table 2), as expected [46]. The absolute, but not the relative, liver mass was higher in HFD mice compared to their respective SD control groups (Table 2). The absolute mass of the pancreas, heart and skeletal muscle was not different from SD, SD-P, HFD and HFD-P groups (Table 2). However, compared to their respective control groups, the relative mass of pancreas, heart and muscle were reduced in the HFD and HFD-P (Table 2). No differences were observed among adipose tissue absolute weights between HFD and HFD-P groups (Table 2). An increase in the relative mass of inguinal and retroperitoneal adipose tissue and BAT were observed in HFD-P mice, compared to HFD mice (Table 2). On the other hand, a decrease in epididymal adipose tissue relative mass was observed between HFD and HFD-P mice (Table 2). No differences in the relative mass of the mesenteric adipose tissue were observed among HFD and HFD-P groups (Table 2).

### 3.2. Pep19 Modulates Insulin Sensitivity and Mass of Endocrine Pancreas in Obese Mice

Compared to the SD group, mice fed an HFD exhibited impaired glucose tolerance (Figure 2A). Chronic administration of Pep19 to mice fed either SD or HFD caused no effects on the overall glucose homeostasis, measured by ipGTT (Figure 2A). Mice fed an HFD presented increased fasting plasma insulin levels compared to control mice fed an SD (Figure 2B). Pep19 administration prevented the increased fasting plasma insulin levels caused by HFD (Figure 2B). In addition, Pep19 administration also prevented increased endocrine pancreas mass caused by HFD (Figure 2D,E). These results suggested that Pep19 administration prevents hyperinsulinemia developed by mice fed an HFD; Pep19 could improve insulin resistance.

### 3.3. Pep 19 Ameliorates Hepatic Steatosis and Inflammation

Histological analyses of liver slices stained with H&E suggested that HFD-fed mice, compared to SD-fed mice, had characteristic fat accumulation, with hepatocytes containing one or more large fat droplets that displace the nucleus to an eccentric position, most prominent in the pericentral (centrilobular) zone (Figure 3A) [54]. Pep19 treatment of HFD-fed mice (HFD-P) largely reduced the size and the number of these characteristic fat droplets accumulated within hepatocytes (Figure 3A). Accordingly, Pep19 treatment (HFD-P) largely attenuated hepatic steatosis and inflammation in HFD-fed mice (HFD; Figure 3B). However, Pep19 treatment of mice fed an HFD (HFD-P) could not reduce the increased levels of liver triacylglycerol (Figure 3C); the levels of plasma triacylglycerol contents were not different among SD and HFD (Figure 3D). Compared to SD-fed mice, HFD-fed mice showed increased levels of plasma cholesterol, which was not attenuated by the treatment with Pep19 (SD-P or HFD-P; Figure 3E). The plasma levels of alanine aminotransferase were increased in mice fed an HFD, compared to that of HFD-fed mice administrated with Pep19 (HFD-P; Figure 3F). Plasma levels of aspartate aminotransferase were not different among mice fed either an SD or HFD (Figure 3G).

### 3.4. Pep 19 Effects on Adipose Tissue Morphology and UCP1 mRNA Expression

Next, the effects of Pep19 treatment on white and BAT morphology were evaluated (Figure 4). Adipocytes area from both inguinal (iWAT) and mesenteric (mWAT) adipose tissues were larger in HFD-fed mice compared to mice fed an SD (Figure 4A,B). The iWAT from the SD-P group presented multilocular brown-fat-like cells (Figure 4A), and increased UCP1 mRNA expression levels (Figure 4B). These data corroborate previous suggestions that Pep19 can induce the conversion of white to brown fat cells (“*brown conversion*”) [21]. Indeed, the iWAT adipocytes from the SD-P group presented a reduced diameter/area, suggesting a direct effect of Pep19 on iWAT (Figure 4C). Conversely, on mWAT, no induction of multilocular brown-fat-like cells (Figure 4A), nor adipocyte diameter reduction (Figure 4D), were observed. Histological analysis of BAT from HFD mice suggested the presence of a large number of white adipocyte-like unilocular cells compared to BAT of SD, SD-P or HFD-P groups (Figure 4A). Thus, Pep19 treatment (HFD-P) prevented BAT to undergo such phenotypic alterations after HFD-induced obesity (HFD). 

### 3.5. Pep19 Treatment Does Not Affect Behavior

Previous studies strongly suggested the lack of central effects of Pep19 in SD-fed mice [21]. Corroborating previous suggestions, Pep19 did not alter the time spent in the open arms (Figure 5A), percent of entries in the open arms (Figure 5B), and the total number of enclosed arms entries (Figure 5C) between HFD and HFD-P mice. Similarly, no significant differences were observed between HFD and HFD-P mice on total immobility time, assessed using a tail suspension test (Figure 5D).

## 4. Discussion

In the present study, the oral administration of Pep19 successfully attenuated body-mass gain and improved the relation between circulating insulin and peripheral insulin sensitivity (disposition index) in a mice model of diet-induced obesity. Pep19 oral administration also ameliorated the hepatic and BAT phenotypes modified by diet-induced obesity. Adipose tissue depots, such as the iWAT in mice, are the depots in which a fraction of the otherwise white-fat-like adipocytes can attain a brown-fat-like appearance and a brown-fat-like thermogenic gene expression profile [55]. In the present study, Pep19 was suggested both to induce morphological changes and to increase UCP1 mRNA levels in the iWAT of SD-fed animals. The increased UCP1 expression in the iWAT is well known as a mechanism that impacts energy balance and glucose uptake, and by which agents induce browning to prevent human obesity and related diseases [56]. The present report suggests that Pep19 oral administration can help to overcome the obesogenic phenotype induced by HFD in a mice model.

Mice models of diet-induced obesity are important tools for understanding the interplay of high-fat Western diets and the development of obesity and associated metabolic diseases [46,57]. The mice model of diet-induced obesity, employed herein, closely mimics the increasing availability of the high-fat/high-density foods in modern society, which are main contributors to the obesity trend in humans [46,57]. Previous report suggests that the oral administration of Pep19 in diet induced male Wistar obese rats remarkably improved metabolic parameters, including a reduction in serum glucose, triacylglycerol and blood pressure, without changing heart rate [21]. Pep19 administration reduced the whole adiposity index and shown increased the number of adipocytes with a significant decrease in their size [21]. In addition, Pep19 increased the expression of UCP1 in specific cells of the inguinal adipose tissue [21]. Herein, the HFD mice model of diet-induced obesity markedly increased body weight and fat depots compared to mice fed an SD. The oral administration of Pep19 to SD-fed mice reduced the relative mass of BAT, as well as of all white adipose tissues evaluated, including inguinal, epididymal, mesenteric and retroperitoneal. Pep19 administration to HFD-fed mice increased the relative mass of inguinal and retroperitoneal adipose tissue and BAT, whereas it decreased epididymal adipose tissue relative mass compared to HFD-fed mice. Possible changes in the total volume of ingested water and the hematocrit analysis, which could add information on the state of hydration of these animals, should be further evaluated. Pep19 oral administration to HFD-fed mice reduced body-mass gain; however, the adipose index was lowered by Pep19 administration only in the SD-fed mice, not in HFD-fed mice. Indeed, animals fed an SD that were orally administrated with Pep19 presented multilocular brown-fat-like cells in the inguinal white adipose tissue and presented increased UCP1 mRNA expression levels, which could corroborate previous suggestions that Pep19 induces the conversion of white to brown fat cells [21]. 

Morphological analysis of the BAT suggested that Pep19 prevents the presence of larger white-like unilocular cells in HFD-fed mice. BAT is one of the most insulin-sensitive tissues [58], and it is essential to glucose homeostasis regulation [59]. BAT activation results in a higher rate of glucose and triglyceride uptake compared with other tissues [60,61]. Obesity results in BAT dysfunction, characterized by decreased fat oxidation and increased lipid deposition, which favors a white-like tissue appearance (whitening) in rodents and humans [62,63]. Both systemic and peripheral CB1-R blockade directly activates BAT, reversing diet-induced obesity and dyslipidemia by selectively enhancing VLDL-TG clearance by metabolically active BAT [64,65]. Pronounced cell infiltration in BAT, in addition to a white adipose-like phenotype, were observed in HFD-fed animals. These obesogenic BAT phenotypes were improved by Pep19 treatment in HFD animals. Moreover, oral administration of Pep19 to HFD-fed mice reduced the absolute mass of the endocrine pancreas and improved the relationship between circulating insulin and peripheral insulin sensitivity. Together, these data corroborate previous suggestions that the oral administration of Pep19 improves several metabolic parameters. Pep19 metabolic improvements seem to occur with great specificity, depending on the metabolic state (i.e., SD vs. HFD) and target tissue (i.e., distinctive adipose tissues, heart and skeletal muscle). Despite the previously described CB1-R inverse agonist (IC_50_ 4.9 × 10^−12^) action of Pep19, further studies are still necessary to address Pep19’s molecular mechanism of action.

Obesity is associated with a state of chronic low-grade inflammation, which may be a pivotal mechanism linking obesity to its metabolic outcomes [66,67]. The inverse agonist of CB1-R rimonabant was previously suggested to increase insulin sensitivity, inducing a shift in macrophage phenotype, from a pro-inflammatory M1 to an anti-inflammatory M2 phenotype [68]. Studies have shown increased expression of inflammatory markers (i.e., TNF-α, IL-1β, and IL-6) by adipose tissue in mice with HFD-induced obesity [69,70,71]. There is also growing evidence that peripheral CB1-R plays critical roles in obesity-induced pro-inflammatory responses, particularly in insulin-target tissues [72,73]. AJ5012, a peripheral CB1-R/CB2-R antagonist, was shown to reduce food intake, weight loss, and increased energy expenditure in diet-induced obese mice [72]. AJ5012 treatment also exhibited effects comparable to CB1-R inverse agonist rimonabant, preventing macrophage infiltration and production of proinflammatory cytokines, which resulted in the suppression of adipose tissue inflammation [72]. Indeed, CB2-R’s contribution to obesity-associated inflammation, insulin resistance, and non-alcoholic fatty liver disease has been previously shown [74]. These data suggested that peripheral CB1-R/CB2-R blockade might break the links between insulin resistance and adipose tissue inflammation. Mechanistically, Pep19 could be acting as an inverse agonist of CB1-R, as previously characterized [21], which could result in the observed improvement of liver inflammation, observed herein. However, it was not in the scope of the present report to investigate the molecular mechanism of action of Pep19.

The liver plays a central role in obesity-related insulin resistance [75] and the steatosis and insulin resistance induced by an HFD depends on the activation of the peripheral CB1 receptors, including those in the liver [76,77,78]. It has been previously demonstrated that pharmacological blockade or inverse agonism of CB1-R improves liver steatosis in animals [79,80] and humans [81]. The CB1-R inverse agonist/antagonist SR141716 decreases CB1-R mRNA expression levels, improving hepatic carbohydrate and lipid metabolism [82]. Conversely, CB1-R mRNA expression levels were increased by arachidonic acid N-hydroxyethyl amide, a CB1-R agonist [82]. Therefore, targeting the endocannabinoid system can be considered a promising anti-obesogenic therapeutic strategy [82]. Consistent with these data, we showed, herein, a reduction in liver steatosis and inflammation in HFD-P animals. It was noteworthy that Pep19 treatment improved ectopic fat accumulation in the liver and reduced alanine aminotransferase activity, suggesting an enhancement in liver function; alanine aminotransferase is present in the liver at much higher concentrations than in other organs, and leakage from the hepatocyte into the blood occurs following hepatocellular injury, being the most widely used clinical biomarker of hepatic health [83]. The reduction in fat in the liver from HFD-P mice is in accordance with studies demonstrating that improving peripheral insulin sensitivity lowers the amount of ectopic fat [84]. One possible explanation for the paradox of Pep19 reducing liver steatosis, without reducing the levels of liver triacylglycerol, could be related to the distribution of fat droplet characteristics of steatosis [54]. These fat droplets mainly accumulate in the pericentral (centrilobular) zone of the liver, while triacylglycerol is distributed throughout the liver tissue [54,85]. Therefore, Pep19 prevents the characteristic fat accumulation of one or more large fat droplets within hepatocytes in the pericentral (centrilobular) zone, without reducing the total contents of triacylglycerol. Further studies should be conducted to further understand the molecular mechanisms behind the important reduction in liver steatosis and inflammation following Pep19 treatment.

Adverse central side effects were observed as the major problems with previous inverse agonists of cannabinoid receptors to treat obesity and metabolic disfunctions [18,19,20]. The action of Pep19 on behavioral components was previously studied, suggesting no effects on the cannabinoid tetrad, depressive nor anxiety-like behaviors [21]. Similarly, results presented herein showed that Pep19 has no behavioral effects (i.e., anxiety- and depressive-like responses). Caloric intake was higher in HFD-fed animals compared to other groups, including the HFD-P, suggesting that Pep19 could induce a reduction in chow consumption, activating thermogenesis, and not through the central nervous system. Indeed, secretin-activated BAT was shown to mediate prandial thermogenesis, inducing satiation [86,87]. Furthermore, the activation of beta-adrenergic receptors in adipocytes stimulates adenylyl cyclase and subsequent cAMP production during non-shivering thermogenesis, which induces lipolysis, and increases UCP1 activity and expression [88]. In this sense, CB1-R inverse agonists, such as Pep19, may contribute to UCP1 activity and expression in brown and white adipocytes by increasing cAMP production, since the CB1-R receptor is associated with Gαi protein. Thus, Pep19 activation of BAT could cause greater metabolic satiety in the animals, reducing chow consumption, without affecting the central nervous system feeding behaviors. These results could corroborate a regulatory role for BAT activation in regulating appetite and energy homeostasis, a mechanism that is also present in healthy humans [87]. 

The protein family of the proton-coupled oligopeptide transporter (POT) consists of four members (i.e., PEPT1 (SLC15A1), PEPT2 (SLC15A2), PHT1 (SLC15A4), PHT2 (SLC15A3)), which could be responsible for the biological membrane transport of small peptides, through an inwardly directed proton gradient and negative membrane potential [28,89,90]. Pept1 gene deletion largely diminished the intestinal uptake and effective permeability of the dipeptide GlySar, and its oral absorption following gastric gavage [91]. PepT1 was also shown to transport larger peptide prodrug molecules, such as valacyclovir, suggesting that PepT1 could transport larger peptides [92]. Indeed, it is also important to mention that in the past 5–6 years, the U.S. Food and Drug Administration (FDA) have authorized the clinical prescription of more than 15 new peptides or peptide-containing molecules [93,94]. These data suggest that oral peptides, such as Pep19, are slowly becoming exciting candidate molecules for improving health and life quality. Overall, the present results suggested that oral administration of Pep19 prevents most of the harmful metabolic alterations related to diet-induced obesity in mice. Clinical studies should be conducted to continue positioning Pep19 as a candidate molecule to improve metabolic parameters, controlling overweight and obesity, without undesired side-effects.

## Figures and Tables

**Figure 1 ijms-23-04082-f001:**
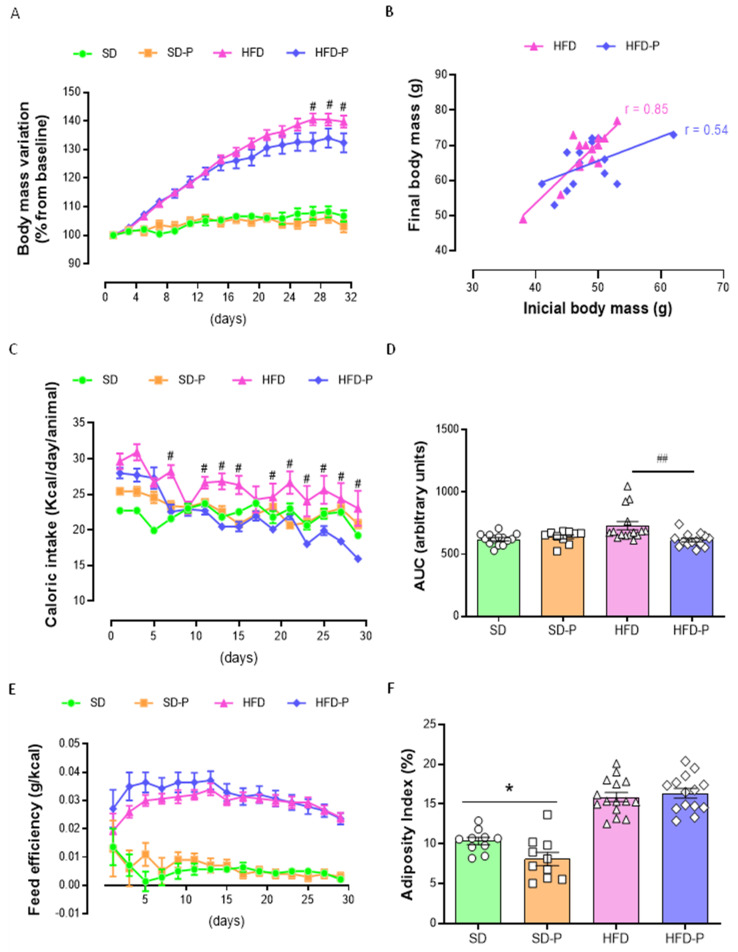
Pep19 treatment effects on body-weight gain, caloric intake, feed efficiency and adiposity index; standard diet (SD), high-fat diet (HFD), SD treated with Pep19 (SD-P), and HFD treated with Pep19 (HFD-P). Pep19 (1 mg/kg) was administered orally by gavage once a day during the morning (08 h–09 h) period. (**A**) Body-weight increase (%). (**B**) Initial and final body-weight correlation curves. (**C**) Caloric intake (Kcal/day/animal). (**D**) Area Under the Curve (AUC) of caloric intake (**C**). (**E**) Feed efficiency curve (g/kcal). (**F**) Adiposity index (%). Data are mean ± SEM. *n* = 10–15 per group. *, *p* < 0.05 SD vs. SD-P; #, *p* < 0.05 HFD vs. HFD-P, ##, *p* < 0.001 HFD vs. HFD-P. Statistical analyses were performed using either Two-way ANOVA with Tukey’s post-hoc test (**A**,**C**,**E**), unpaired *t*-test (**D**,**F**) or correlation analyses with Pearson r calculation (**B**).

**Figure 2 ijms-23-04082-f002:**
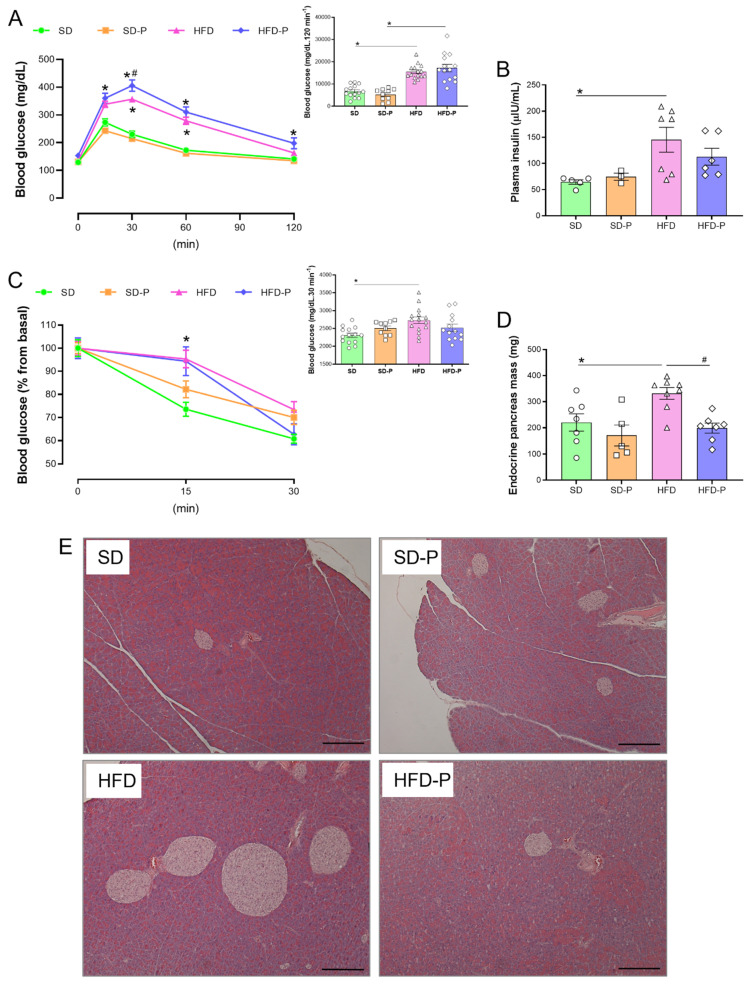
Pep19 oral treatment prevents HFD-induced hyperinsulinemia and increase in the absolute mass of endocrine pancreas. (**A**) Glucose tolerance test, graphic and area under the curve (AUC; inset). (**B**) Plasma insulin (µIU/mL). (**C**) Insulin tolerance test, graphic and AUC (inset). (**D**) The absolute mass of the endocrine pancreas. (**E**) Representative pictures from pancreas sliced sections stained with hematoxylin and eosin (H&E). Scale bar, 200 μm. Data expressed as mean ± SEM. *n* = 4–15 per group. *, *p* < 0.05 between SD vs. HFD; #, *p* < 0.05, HFD vs. HFD-P. Statistical analyses were performed using either Two-way ANOVA with Tukey’s post-hoc test (**A**,**C**) or unpaired *t*-test (insets from (**A**–**D**)).

**Figure 3 ijms-23-04082-f003:**
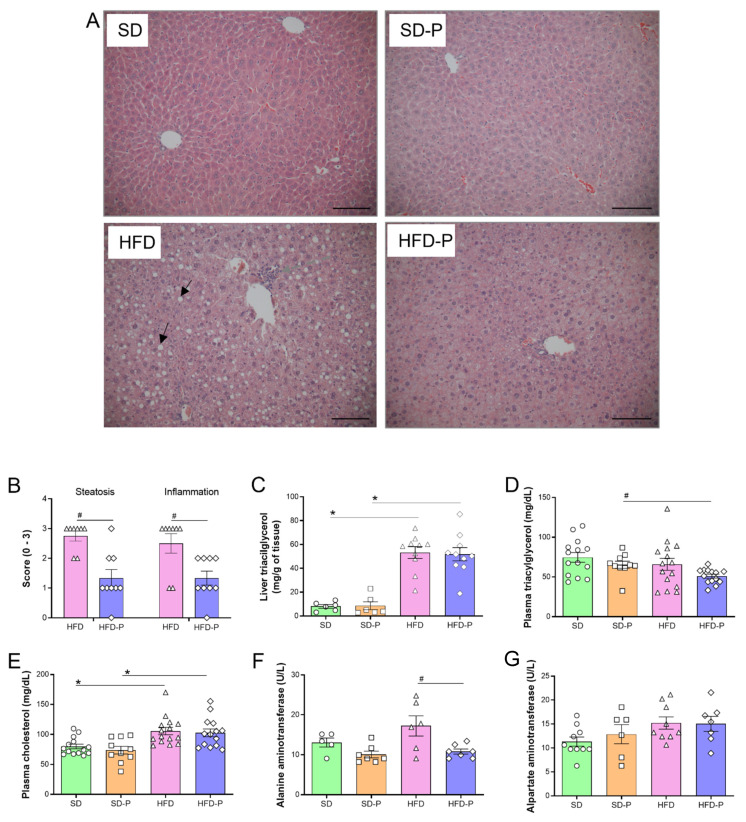
Pep 19 ameliorates hepatic steatosis and inflammation. (**A**) Representative pictures from liver sliced sections stained with hematoxylin and eosin (H&E); the black arrows show typical macro vesicular steatosis and the white arrows show inflammatory foci. Scale bar, 200 μm. (**B**) Liver steatosis was scored and the severity was graded as detailed in the Materials and Methods section. Inflammation was evaluated by counting the number of inflammatory foci per field, and was scored as detailed in the Materials and Methods section. Both steatosis and inflammation were evaluated at 100 × magnification. (**C**–**G**) biochemical parameters: (**C**) liver triacylglycerol; (**D**) plasma triacylglycerol; (**E**) plasma cholesterol; (**F**) plasma aspartate aminotransferase activity; (**G**) plasma alanine aminotransferase activity. Data were expressed as: (**B**), median with interquartile range; (**C**–**G**), mean ± SEM. *n* = 5–15 per group. *, *p* < 0.05; between SD vs HFD or SD-P vs. HFD-P; #, *p* < 0.05; between HFD vs. HFD-P. Statistical analyses were performed using either non-parametric Mann–Whitney *t*-test (**B**) or unpaired *t*-test (**C**–**G**).

**Figure 4 ijms-23-04082-f004:**
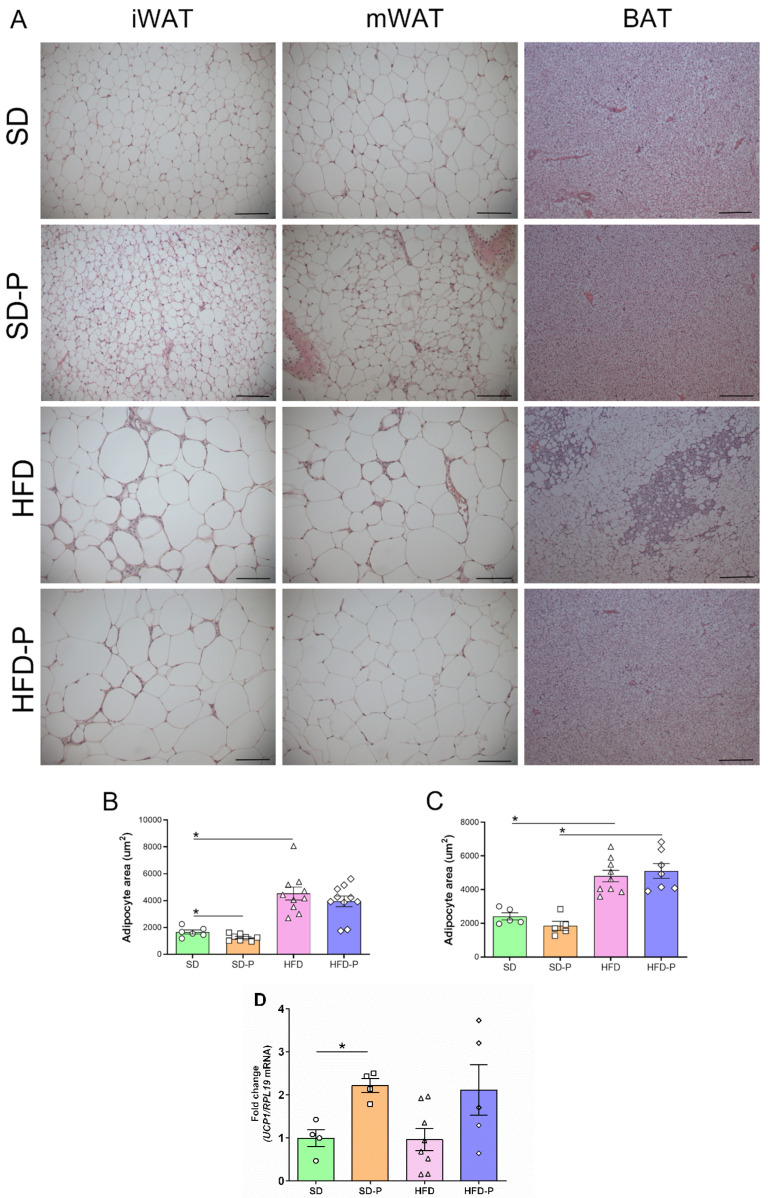
Effects of Pep19 treatment on adipose tissue morphology. (**A**) H&E-stained representative images of inguinal white adipose tissue (iWAT), mesenteric white adipose tissue (mWAT) or brown adipose tissue (BAT), from animals fed either standard diet (SD) or high-fat diet (HFD). Pep19 (1 mg/Kg) was orally administered to mice that received either SD (SD-P) or HFD (HFD-P). (**B**,**C**), quantitative analyses of the white adipocytes area from H&E-stained slices from iWAT (**C**) or mWAT (**D**) adipose tissues from animals fed either SD or HFD, orally administrated with either saline vehicle (SD or HFD) or Pep19 (1 mg/Kg; SD-P or HFD-P). (**D**) UCP1 mRNA expression level was investigated on the iWAT from animals fed either SD or HFD, orally administrated with either saline vehicle (SD or HFD) or Pep19 (1 mg/Kg; SD-P or HFD-P). Measurements were performed on at least 15 different adipocytes from each of the 6 fields analyzed per H&E-stained slices from each animal. Scale bar, 100 μm. Data were expressed as mean ± SEM. *n* = 4–15 per group. *, *p* < 0.05 between SD vs. HFD, non-administered or administered with Pep19 as indicated (SD-P or HFD-P, respectively). Statistical analyses were performed using unpaired *t*-test followed by ad-hoc Tukey’s test (**B**–**D**), using GraphPad Prism 8 software.

**Figure 5 ijms-23-04082-f005:**
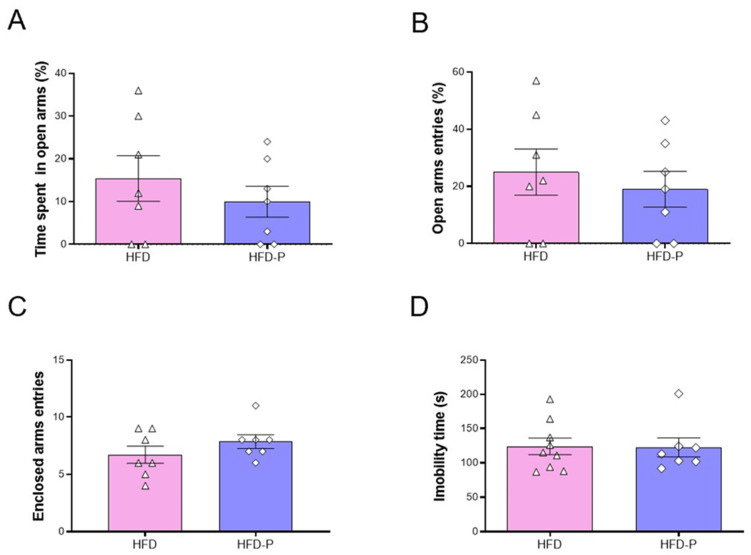
Comparative behavioral analyses of HFD-fed animals following oral administration of either saline vehicle (HFD) or Pep19 (1 mg/Kg; HFD-P). (**A**–**C**) Analysis of the effect of Pep19 on anxiety evaluated through the elevated plus-maze test. (**A**) Time spent in open arms (%). (**B**) Open arms entries (%). (**C**) Number of entries into Enclosed arms. (**D**) Analysis of the effect of Pep19 on depressive-like behavior evaluated by tail suspension test. Immobility time (s). Data expressed as mean ± SEM. *n* = 7–8 animals per group. No significant differences (*p* < 0.05) were observed among different groups. Statistical analyses were performed using unpaired *t*-test (**A**–**D**).

**Table 1 ijms-23-04082-t001:** Nutritional composition of standard diet (SD) and high-fat diet (HFD).

Ingredients	SD		HFD	
	g/Kg	kcal	g/Kg	Kcal
Maize starch	415	1660	14.3	57.2
Soybean meal	305	1281	410	1722
Sucrose	80	320	80	320
Maltodextrin	70	280	70	280
Lard	0	0	302	2718
Soy oil	0	0	0	0
Soy fatty acid	50	350	50	350
Cellulose	31.7	0	25.4	0
L-cysteine	1.8	7.2	1.8	7.2
Choline chloride	1.5	0	1.5	0
Butylhydroxytoluene	0.014	0	0.028	0
Vitamin’s mix	10.0	40	10	40
Total	-	3938	-	5494

**Table 2 ijms-23-04082-t002:** Tissue weights from experimental groups.

	SD	SD-P	HFD	HFD-P
**Absolute Mass (g)**				
Inguinal adipose tissue	0.93 ± 0.07	0.71 ± 0.10	1.99 ± 0.15 ^a^	2.13 ± 0.16 ^b^
Epididymal adipose tissue	1.66 ± 0.11	1.32 ± 0.15	3.48 ± 0.19 ^a^	3.14 ± 0.18 ^b^
Mesenteric adipose tissue	1.18 ± 0.05	0.75 ± 0.07 *	2.71 ± 0.18 ^a^	2.77 ± 0.25 ^b^
Retroperitoneal adipose tissue	1.29 ± 0.07	0.89 ± 0.14 *	2.10 ± 0.16 ^a^	2.13 ± 0.13 ^b^
Brown adipose tissue	0.29 ± 0.01	0.20 ± 0.02 *	0.40 ± 0.02 ^a^	0.45 ± 0.03 ^b^
Liver	1.76 ± 0.05	1.64 ± 0.06	2.46 ± 0.14 ^a^	2.46 ± 0.16 ^b^
Pancreas	0.31 ± 0.01	0.29 ± 0.01	0.26 ± 0.01	0.26 ± 0.02
Heart	0.16 ± 0.01	0.16 ± 0.01	0.17 ± 0.01	0.17 ± 0.01
Muscle	0.25 ± 0.01	0.25 ± 0.01	0.24 ± 0.01	0.23 ± 0.01
**Relative Mass (%)**				
Inguinal adipose tissue	1.79 ± 0.14	1.44 ± 0.20 *	2.93 ± 0.16	3.22 ± 0.18 ^##, b^
Epididymal adipose tissue	3.17 ± 0.21	2.68 ± 0.28 *	5.19 ± 0.23	4.84 ± 0.26 ^##, b^
Mesenteric adipose tissue	2.26 ± 0.09	1.54 ± 0.12 **	4.00 ± 0.19	4.18 ± 0.30 ^b^
Retroperitoneal adipose tissue	2.46 ± 0.12	1.80 ± 0.27 **	3.10 ± 0.19	3.24 ± 0.16 ^#, b^
Brown adipose tissue	0.56 ± 0.03	0.42 ± 0.05 *	0.59 ± 0.03	0.69 ± 0.04 ^##, b^
Liver	3.37 ± 0.06	3.39 ± 0.10	3.66 ± 0.16	3.76 ± 0.18
Pancreas	0.59 ± 0.03	0.61 ± 0.03 ^a^	0.40 ± 0.02 ^a^	0.41 ± 0.03 ^b^
Heart	0.31 ± 0.01	0.33 ± 0.01 *	0.26 ± 0.01 ^a^	0.26 ± 0.01 ^b^
Muscle	0.48 ± 0.01	0.52 ± 0.01 *	0.36 ± 0.02 ^a^	0.35 ± 0.02 ^b^

SD standard diet, SD-P standard diet treated with Pep19 (1 mg/kg), HFD high-fat diet, HFD-P high-fat diet treated with Pep19 (1 mg/kg). Data are mean ± SEM. *n* = 10–15 per group. *, *p* < 0.05 SD-P vs. SD; **, *p* < 0.001 SD-P vs. SD; #, *p* < 0.05 HFD-P vs. HFD; ##, *p* < 0.001 HFD-P vs. HFD; ^a^, *p* < 0.05 SD vs. HFD; ^b^, SD-P vs. HFD-P. Statistical analyses were performed using unpaired *t*-test.

## Data Availability

Not applicable.

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
