# Peer review of "Pep19 Has a Positive Effect on Insulin Sensitivity and Ameliorates Both Hepatic and Adipose Tissue Phenotype of Diet-Induced Obese Mice"

_ijms, 2022, doi:10.3390/ijms23084082_

Round 1
Reviewer 1 Report
he topic of manuscript entitled "Pep19 treatment has a positive effect on insulin sensitivity and ameliorates both hepatic and adipose tissue phenotype of diet-induced obese mice" is interesting, and the manuscript is well organized. But the manuscript need to be revised before accepted for publication.
- Delete "treatment" in the title;
- Authors are not ended;
- The references format should be revised.
Author Response
Reviewer #1
The topic of manuscript entitled "Pep19 treatment has a positive effect on insulin sensitivity and ameliorates both hepatic and adipose tissue phenotype of diet-induced obese mice" is interesting, and the manuscript is well organized. But the manuscript needs to be revised before accepted for publication.
Delete "treatment" in the title;
Authors are not ended;
The references format should be revised.
Rebuttal letter
We would like to thank the reviewer for his/her meaningful revision and suggestions of our current manuscript. Accordingly, we have addressed all the questions appropriately in the revised version of the manuscript; all editions made were labeled for reviewers` consultation.
Point-to-point questions and answers:
1) Delete "treatment" in the title;
Answer: thank you for your suggestion. “Treatment” was deleted from the title.
2) Authors are not ended;
Answer: thank you for your observation. “and” was added to the appropriate position.
3) The references format should be revised.
Answer: thank you for your observation. We have double checked the format of all references using the MDPI instructions for Endnote version 9.
Reviewer 2 Report
To Author
In the present manuscript, Silvério et al. investigated the anti-obesogenic effect of natural peptide DIIADDEPLT (Pep19). The authors used high-fat diet-induced mice as an obesity model. Both the SD and HFD mice were treated with Pep19(1mg/kg) for 30 days. The authors observed that the Pep19 administration resulted in the browning on inguinal white adipose cells along with improved peripheral insulin sensitivity and reduced liver inflammation without any adverse central nervous system effects. Body mass and food intake were measured every two days. The metabolic parameters were analysed using Glucose tolerance and insulin sensitivity assessment. The Histological analysis of pancreas and adipose tissues was conducted to understand the morphological alterations. Real-time PCR was conducted to evaluate the UCP1 mRNA expression. The tail suspension and plus-maze were conducted to analyse depressive and anxiety-like behaviors on experimental obese mice.
- The manuscript is poorly written. In materials and method 2.2. Experimental design and groups Line 131 to 134 is confusing and should be restructured. Also, the discussion section line (417 to 421) looks incomplete.
- Bat genes PGC-1α, Cidea, and Plin5 are associated with the browning of adipose tissue. In the present study author only examined the gene expression of What’s the rationale behind selecting Ucp1 for gene expression analysis?
- In the present study, the author mentioned that the obese mice treated with Pep 19 didn’t show any changes in behaviors. Most of the CB1-R blocking drugs showed a high Blood-brain barrier permeability. What about the BBB permeability of the Pep 19?
Minor comments
- In Table 1. Nutritional composition of SD and HFD Check soyabean meal Kcal 17220 or 1722
- In figure 2B y-axis Font is not clear
- Figure legend in figure 4 does not match
- Line 490 font size is different
Author Response
Rebuttal letter reviewer #2
In the present manuscript, Silvério et al. investigated the anti-obesogenic effect of natural peptide DIIADDEPLT (Pep19). The authors used high-fat diet-induced mice as an obesity model. Both the SD and HFD mice were treated with Pep19(1mg/kg) for 30 days. The authors observed that the Pep19 administration resulted in the browning on inguinal white adipose cells along with improved peripheral insulin sensitivity and reduced liver inflammation without any adverse central nervous system effects. Body mass and food intake were measured every two days. The metabolic parameters were analysed using Glucose tolerance and insulin sensitivity assessment. The Histological analysis of pancreas and adipose tissues was conducted to understand the morphological alterations. Real-time PCR was conducted to evaluate the UCP1 mRNA expression. The tail suspension and plus-maze were conducted to analyse depressive and anxiety-like behaviors on experimental obese mice.
The manuscript is poorly written. In materials and method 2.2. Experimental design and groups Line 131 to 134 is confusing and should be restructured. Also, the discussion section line (417 to 421) looks incomplete.
Answer: thank you for your observation. We have double checked the manuscript for English grammar and typos. We have addressed all your specific comments (i.e. Experimental design and groups Line 131 and discussion section line (417 to 421), and additionally have also tried to improve the manuscript overall. All changes made on the manuscript text were appropriately labeled in the text for authors consultation.
Bat genes PGC-1α, Cidea, and Plin5 are associated with the browning of adipose tissue. In the present study author only examined the gene expression of What’s the rationale behind selecting Ucp1 for gene expression analysis?
Answer: thank you for your important observation. The rationale behind evaluating the expression of UCP1 was to perform comparative analyses, based on our previous publication showing that Pep19 activates the UCP1 expression in a Wistar rat model of obesity [1]. In the current manuscript, we have observed that Pep19 reduced the size of adipocytes from inguinal adipose tissue (Figure 4), similar to what was previously observed in the Wistar rat model [1]. One of the possible explanations for the decrease of adipose cells size was the reduction of fat content, due to an increased metabolic activity of UCP1 within the inguinal adipose tissue. Measuring the expression of UCP1 we could corroborate our suggestion that an increase of thermogenic activity in the inguinal adipose tissue could be related to the adipose size decrease upon Pep19 treatment.
In the present study, the author mentioned that the obese mice treated with Pep 19 didn’t show any changes in behaviors. Most of the CB1-R blocking drugs showed a high Blood-brain barrier permeability. What about the BBB permeability of the Pep 19?
Answer: thank you for your intriguing and interesting observation. There is no information in the literature regarding the presence of Pep19 in brain tissue following oral or intraperitoneal administration to rats or mice. However, we would like to argue that previous reports shown that the peptide hemopressin [2-4], without crossing the blood brain barrier [5], could activate the central nervous system reducing food intake and causing anxiogenic-like behavior in mice [1,5-7]. Therefore, we concluded that for Pep19 the most relevant experiments to be performed were the behavior tests, which could identify or not a possible central nervous system effect, which in turn could suggest that Pep19 could have possible clinical advantages compared to rimonabant. The direct effects of Pep19 on mice behavior was previously investigated intensely, using the cannabinoid tetrad following both acute and chronic administration of Pep19 [1]. Herein, we have used the elevated plus-maze and the tail suspension tests to corroborate previous observation that Pep19 lacks behavioral effects following oral administration to mice.
Minor comments
1) In Table 1. Nutritional composition of SD and HFD Check soyabean meal Kcal 17220 or 1722
Answer: thank you so much for your observation. We have made the correction that soybean meal has 1722 Kcal.
2) In figure 2B y-axis Font is not clear
Answer: thank you for your observation. We have checked Figure 2B y-axis Font and could not identify the unclear Font.
3) Figure legend in figure 4 does not match.
Answer: thank you for your observation. We have made the appropriate corrections to the legend of Figure 4.
4) Line 490 font size is different
Answer: thank you for your observation. We have made the appropriate corrections to line 490 font size.
- Reckziegel, P.; Festuccia, W.T.; Britto, L.R.G.; Jang, K.L.L.; Romao, C.M.; Heimann, J.C.; Fogaca, M.V.; Rodrigues, N.S.; Silva, N.R.; Guimaraes, F.S., et al. A novel peptide that improves metabolic parameters without adverse central nervous system effects. Sci Rep 2017, 7, 14781, doi:10.1038/s41598-017-13690-9.
- Heimann, A.S.; Gomes, I.; Dale, C.S.; Pagano, R.L.; Gupta, A.; de Souza, L.L.; Luchessi, A.D.; Castro, L.M.; Giorgi, R.; Rioli, V., et al. Hemopressin is an inverse agonist of CB1 cannabinoid receptors. Proc Natl Acad Sci U S A 2007, 104, 20588-20593, doi:10.1073/pnas.0706980105.
- Rioli, V.; Gozzo, F.C.; Heimann, A.S.; Linardi, A.; Krieger, J.E.; Shida, C.S.; Almeida, P.C.; Hyslop, S.; Eberlin, M.N.; Ferro, E.S. Novel natural peptide substrates for endopeptidase 24.15, neurolysin, and angiotensin-converting enzyme. J Biol Chem 2003, 278, 8547-8555, doi:10.1074/jbc.M212030200.
- Heimann, A.S.; Dale, C.S.; Guimarães, F.S.; Reis, R.A.M.; Navon, A.; Shmuelov, M.A.; Rioli, V.; Gomes, I.; Devi, L.L.; Ferro, E.S. Hemopressin as a breakthrough for the cannabinoid field. Neuropharmacology 2021, 183, 108406, doi:10.1016/j.neuropharm.2020.108406.
- Fogaça, M.V.; Sonego, A.B.; Rioli, V.; Gozzo, F.C.; Dale, C.S.; Ferro, E.S.; Guimarães, F.S. Anxiogenic-like effects induced by hemopressin in rats. Pharmacol Biochem Behav 2015, 129, 7-13, doi:10.1016/j.pbb.2014.11.013.
- Dodd, G.T.; Worth, A.A.; Hodkinson, D.J.; Srivastava, R.K.; Lutz, B.; Williams, S.R.; Luckman, S.M. Central functional response to the novel peptide cannabinoid, hemopressin. Neuropharmacology 2013, 71, 27-36, doi:10.1016/j.neuropharm.2013.03.007.
- Dodd, G.T.; Mancini, G.; Lutz, B.; Luckman, S.M. The peptide hemopressin acts through CB1 cannabinoid receptors to reduce food intake in rats and mice. J Neurosci 2010, 30, 7369-7376, doi:10.1523/jneurosci.5455-09.2010.
Round 2
Reviewer 2 Report
Authors answered the quires raised by the reviewer and made necessary corrections in the figure legends tables. However, the corrections made in the discussion section line no 421-425 is not satisfactory. Authors could improve the sentence like. “Previous report suggests that the oral administration of Pep19 in diet induced male Wistar obese rats remarkably improved metabolic parameters, including a reduction in serum glucose, triacylglycerol and blood pressure, without changing heart rate (21)” Also in Line no 426-428. “Pep19 was also previously shown to reduce the whole adiposity index and to increase the number of adipocytes with a significant decrease in their size” . Can be changed to “Pep19 administration reduced the whole adiposity index and shown increased the number of adipocytes with a significant decrease in their size”